# Validating the 2-minute walk test MCID for subacute stroke patients: A Pakistani multicenter cohort analysis

Muslim Khan[1], Hafiz Shehzad Muzamil[2], Ahmad M Osailan [3], Ayman Abdullah Alhammad[4], Samiullah Khan[5], Mazhar Mushtaq[6], Engy BadrEldin Saleh Moustafa[7,8], Theeb Naif S. Alsalem[9], Edward Muteesasira[10]*

**1** Department of Rehabilitation Sciences, Iqra National University, Swat, Pakistan, **2** Department of Human Nutrition and Technology, Faculty of Allied Health Sciences, Superior University, Lahore, Pakistan, **3** Department of Health and Rehabilitation Sciences, College of applied Medical Sciences, Prince Sattam bin Abdulaziz University, Alkharj, Saudi Arabia, **4** College of Medical Rehabilitation Sciences, Taibah University, Saudi Arabia, **5** Department of Rehabilitation Sciences, Iqra National University, Swat, Pakistan, **6** Sulaiman Al Rajhi University, Saudi Arabia, **7** Department of Physical Therapy for Neurology and Neurosurgery, Faculty of Physical Therapy, Cairo University, Cairo, Egypt, **8** Department of Physiotherapy, College of Health Sciences, American University in the Emirates, Dubai, United Arab Emirates, **9** King Abdulaziz Medical City, National Guard Health Affairs, Riyadh, Saudi Arabia, **10** Department of Physiotherapy, Faculty of Health Sciences, Mbarara University of Science and Technology, Mbarara, Uganda

* muteesaedward93@gmail.com

## Abstract

### Background

The 2-Minute Walk Test (2MWT) is a simple and reliable test used by clinicians to evaluate mobility gains in stroke populations.

### Objective

To determine the minimally clinically important difference (MCID) for the 2-Minute Walk Test (2MWT) in subacute stroke populations.

### Design

This was a prospective cohort study.

### Setting

The study was carried out across seven rehabilitation centers in Pakistan.

### Participants

150 adults (<180 days post-stroke),with a confirmed ischemic or hemorrhagic stroke, able to walk at least 10 meters, and Mini-Mental State Examination score ≥24.

**Data availability statement:** The data underlying the results presented in the study are available from Figshare: https://figshare.com/articles/figure/_b_Determining_the_Minimally_Clinically_Important_Difference_of_the_2-Minute_Walk_Test_in_Individuals_with_Subacute_Stroke_A_Multicenter_Cohort_Study_b_/29484728.

**Funding:** The author(s) received no specific funding for this work.

**Competing interests:** The authors have declared that no competing interests exists.

**Abbreviations:** 2MWT: 2-Minute Walk Test; MCID: Minimally Clinically Important Difference; ABC_Gait: Activities-specific Balance Confidence gait subscale; GPE: Global Perceived Effect; MBI: Modified Barthel Index.

## Interventions

Participants underwent 6–8 weeks of standard rehabilitation, including physical therapy (gait and balance training) and, in some centers, robotic gait training. Interventions varied by center but followed standardized gait and balance goals.

## Main outcome measures

The 2MWT was the primary outcome measure whereas, the Activities-specific Balance Confidence gait subscale (ABC_Gait), Global Perceived Effect (GPE) scale, and Modified Barthel Index (MBI) were the secondary outcome measures. The MCID was estimated using an anchor-based approach (ABC_Gait ≥10% improvement), validated by distribution-based methods.

## Results

The 2MWT MCID was 33 meters (95% CI: 30–36 meters), with 87% sensitivity and 82% specificity. A strong correlation was observed between 2MWT and ABC_Gait change scores (r = 0.68, p < 0.001). The mean 2MWT distance improved from 62.5±38.4 meters to 98.7±42.1 meters (p < 0.001). Subgroup analyses showed that MCID was consistent across age groups, stroke types, and intervention modalities.

## Conclusion

The 33 meters MCID for the 2MWT is a reliable and patient-centered benchmark for assessing mobility gains in subacute stroke populations. Validation in chronic stroke populations and exploration of emerging interventions like virtual reality (VR) is needed to extend the 2MWT MCID's applicability.

## Introduction

Globally, stroke is a leading cause of disability impairing the mobility of 70–80% of survivors thus hampering their independence and overall quality of life [1,2]. With over 12 million stroke cases annually worldwide, practical rehabilitation approaches like the 2MWT (2-Minute Walk Test) are necessary to assess meaningful mobility gains achieved with subacute rehabilitation, a critical step towards functional independence. Effective rehabilitation during the subacute phase (<180 days post-stroke) is crucial as neuroplasticity peaks thus providing an opportunity for motor recovery [3–5].

The 2MWT, measuring distance walked in two minutes, is a practical alternative to the 6MWT (6-Minute Walk Test) with established reliability in stroke populations [6,7]. However, its lack of a MCID (Minimally Clinically Important Difference) limits its ability to guide clinical decisions, unlike the 6MWT which has a known MCID of 50 meters [8].

The 2MWT MCID is critical for interpreting clinical significance yet few studies [1,9] have explored it for short-duration walking tests in stroke with most literature having

tested it in other conditions like chronic obstructive pulmonary disease(COPD) [10–12]. Recent research emphasizes the integration of objective and subjective anchors such as the ABC_Gait (Activities-specific Balance Confidence gait sub-scale), to ensure patient-centered outcomes [13]. For instance, studies on Parkinson's disease and spinal cord injury have reported MCIDs for walking tests ranging from 15–30 meters, highlighting the need for condition-specific benchmarks due to varying motor impairments. In Pakistan, limited access to advanced rehabilitation technologies underscores the need for practical tools like the 2MWT [14,15].

This study aimed to: (I) determine the minimally clinically significant difference (MCID) of the 2MWT in patients with subacute stroke using an anchor-based approach,(II) evaluate the correlation between 2MWT improvements and Activities-specific balance confidence gait subscale (ABC_Gait) and global perceived effect (GPE),and lastly (III) assess the applicability of the 2MWT MCID as a patient-centered tool for rehabilitation outcomes in resource-limited settings.

## Methods

### Study design

This was a prospective, longitudinal cohort study conducted across seven rehabilitation centers in Pakistan between January 2023 and June 2025.The study adhered to STROBE (Strengthening the Reporting of Observational studies in Epidemiology) guidelines for observational studies.

### Setting and centers

The seven rehabilitation centers (Table 1) were selected based on the following criteria;

(I) Location in major urban areas across different regions of Pakistan, (II) At least 5 years of experience providing stroke rehabilitation services, (III) Staffed by at least two neurological physical therapists with postgraduate qualifications, (IV) Equipped with standardized gait assessment tools (e.g., 30-meter walkway),and lastly (V) Capacity to enroll at least 20 eligible participants over the study period.

This inclusion criteria ensured that the centers had adequate expertise and resources to implement the study protocol consistently. Centers were recruited through the Pakistan Rehabilitation network. While specific interventions varied based on individual patient needs, all centers adhered to the following core components;

(I) Minimum of 45 minutes of physical therapy 3 times per week consisting of task-specific gait and balance training (e.g., treadmill walking, obstacle courses)

(II) Progressive strength training for lower extremities inclusive of functional mobility practice (e.g., transfers, stair climbing)

**Table 1. Centers.**

| Center | Physical Therapy Components | Technology-Based Interventions | Session Frequency | Duration (Weeks) | Center |
|--------|-----------------------------|--------------------------------|-------------------|------------------|--------|
| A | Gait training, balance exercises | Robotic gait training (20% of participants) | 2–3/week | 6–8 | A |
| B | Gait training, balance exercises | Robotic gait training (25% of participants) | 2–3/week | 6–8 | B |
| C | Gait training, balance exercises | None | 2–3/week | 6–8 | C |
| D | Gait training, balance exercises | Robotic gait training (30% of participants) | 2–3/week | 6–8 | D |
| E | Gait training, balance exercises | None | 2–3/week | 6–8 | E |
| F | Gait training, balance exercises | None | 2–3/week | 6–8 | F |
| G | Gait training, balance exercises | None | 2–3/week | 6–8 | G |

Table 1 summarizes the intervention protocols implemented at each of the seven rehabilitation centers.

## Participants

We recruited 150 adults (aged 18 or older), < 180 days after a confirmed ischemic or hemorrhagic stroke. They were admitted to inpatient or outpatient rehabilitation settings covered comprehensively by the Pakistani national health service.

## Inclusion criteria

(I)    Ability to walk at least 10 meters (with or without any assistive device),

(II)   Shared rehabilitation goals aimed at improving gait and balance, and lastly,

(III)  Mini-Mental State Examination score ≥24 [16].

## Exclusion criteria

(I)    Severe cognitive or communication impairments preventing test comprehension,

(II)   Age under 18 and therefore unable to provide informed consent,

(III)  Comorbidities (e.g., severe cardiopulmonary disease) that markedly limited mobility, and lastly,

(IV)  A participant in other interventional studies.

**Sample size determination.** The initial screening pool was calculated using the Fleiss' formula [17] resulting into 431 individuals. that were screened for eligibility across the seven centers.

$$N = \frac{(Z\propto + Z\beta)(Z\propto + Z\beta) * (P0(1-P0) + P1(1-P1))}{(P1-P0)(P1-P0)}$$

Whereby;
"N" is the initial screening pool, P0 is the significance level estimated at 5% (0.05), P1 is the exposed proportion in exposed group (P1 = P0*risk ratio) = 0.05*2 = 0.1, Z alpha is the 95% confidence value of 95% estimated at 1.96, Z beta is the normal deviation for the desired power of 80% estimated at 0.84
Applying the above values;

$$N = \frac{(1.96 + 0.84)(1.96 + 0.84)) * (0.05(1-0.05) + 0.1(1-0.1))}{(0.1-0.05)(0.1-0.05)}$$

$$N = 431.2$$

Therefore, the initial screening pool was 431 individuals.

The initial screening pool of 431 individuals was subjected to our inclusion/exclusion criteria resulting into 150 participants that met the inclusion criteria and completed the study as shown in (Fig 1). A power analysis confirmed that n = 150 provides 80% power at α = 0.05 to detect the MCID ensuring the study is robustly powered despite the exclusion of non-eligible screened individuals.

## Inclusivity in global research

Additional information regarding the ethical, cultural, and scientific considerations specific to inclusivity in global research is included in the Supporting Information (S1 Checklist).

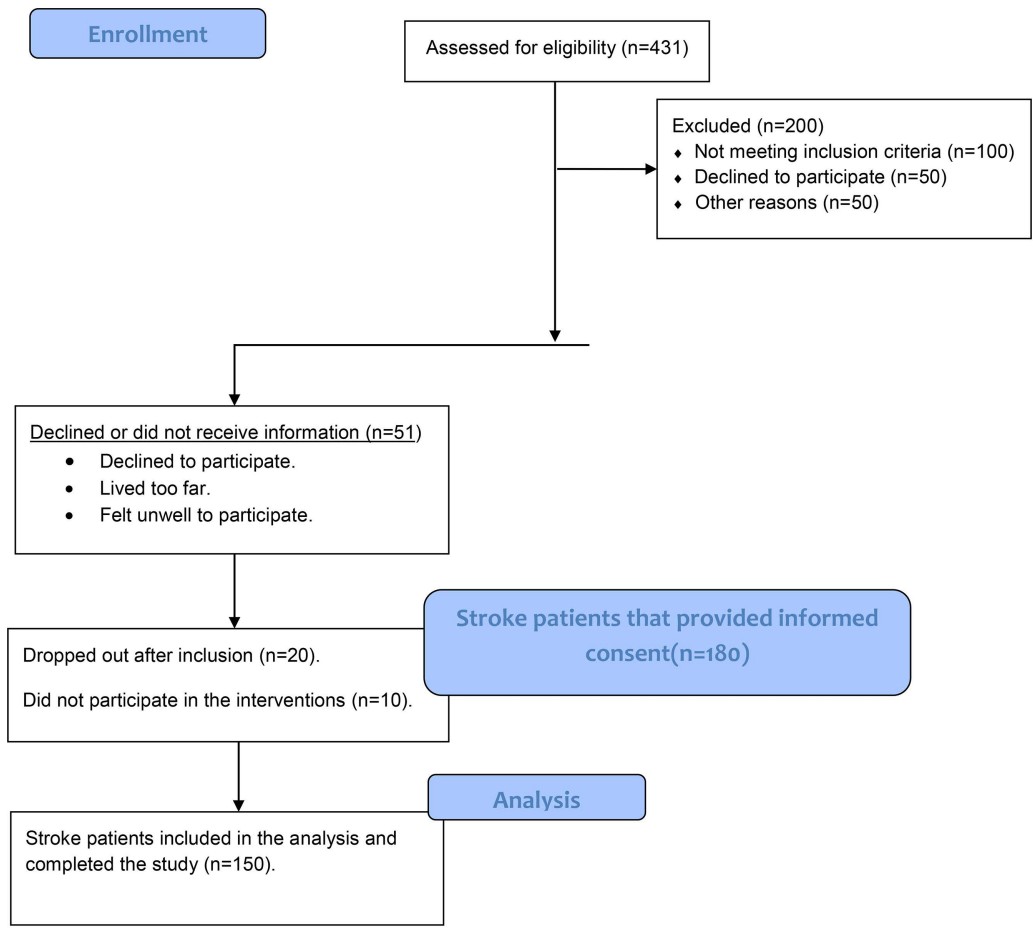

**Fig 1. Participant flow chart.**

**Interventions and data collection.** Participants received personalized rehabilitation programs over 6–8 weeks (12–15 sessions) and included the following interventions;

(I) Physical therapy (e.g., gait training, balance exercises),(II) Task-specific activities (e.g., treadmill walking with body-weight support, overground walking with obstacle navigation) and lastly, and lastly,(III) Technology-based interventions where available at three centers(A,B and D) as shown in Table 1 and included robotic gait training. To ensure consistency across centers, rehabilitation protocols were guided by standardized gait and balance goals carried out through specific interventions which varied based on center resources and patient needs.

**Standardization was achieved through.** (I) a physiotherapist training manual that specified core gait and balance exercises, (II) biweekly audits to ensure adherence across centers, and lastly, (III) a minimum of 45 minutes per session focused on gait/balance training. All assessments were conducted indoors on a standardized 30-meter walkway with a non-slip surface.

Temperature was maintained at 20–25°C across all centers. Trained researchers, blinded to the rehabilitation process to minimize bias, assessed participants at two points; **T0**: Baseline (pre-intervention) and **T1**: Post-intervention (6–8 weeks). In order to maintain consistency, the same training procedures such as covering assessor training and the use of a 30-meter walking track following the 2MWT methodology, were applied across the different centers. Assessors were

blinded to baseline measurements and intervention details. Randomization was not performed due to the nature of the observational cohort design while allocation concealment was not applicable.

## Outcome measures

**Primary outcome.** 2-Minute Walk Test (2MWT) [18]; participants walked as far as possible in 2 minutes along the 30-meter walkway. The average distance of two trials was recorded in meters.

**Secondary outcomes.** (I) Activities-specific Balance Confidence (ABC) Scale [19], (II) a shortened version focusing on gait-related items (ABC_Gait) [20] assessed confidence in maintaining balance during walking tasks (score range: 0–100), (III) Global Perceived Effect (GPE) [21], (IV) Participants global perceived effect (GPE_pwST) [22], (V) Physiotherapists global perceived effect (GPE_PT) [23], which assessed perceived changes in walking ability using a 7-point Likert scale (1 = much worse, 7 = much better) and lastly, (VI) Modified Barthel Index (MBI) [24], which assessed functional independence in activities of daily living.

## Data analysis

According to the criteria of normality, we utilized;(I) paired t-tests to compare pre- and post-intervention scores, logistic regression to examine associations between 2MWT improvements and baseline data (age, sex, stroke severity, baseline 2MWT) and adjusting for confounders, (II) calculation of standard error of measurement (SEM) and effect size using distribution-based methods,(III) multiple imputation to address missing data (8% of cases),(IV) Sensitivity analyses inclusive of complete-case analysis to confirm consistency of MCID estimates, and lastly,(V) subgroup comparisons (e.g., robotic vs. manual gait training, age groups, stroke types)

All statistical analyses were conducted using SPSS v.28.0 (IBM Corp., Armonk, NY), with significance set at p < 0.05.

A 10% improvement in ABC_Gait was selected as the anchor because it represents a threshold beyond the standard error of measurement (SEM) for this scale, ensuring that the identified change is clinically meaningful and perceptible to the patient as highlighted by prior studies [1,13,25]. The optimal MCID cutoff, based on Receiver Operating Characteristic (ROC) curves, indicated maximum sensitivity and specificity (area under the curve [AUC]> 0.89).Paired t-tests were performed every three weeks on 2MWT,ABC, ABC_Gait, and MBI scores.

A sensitivity analysis using complete-case analysis confirmed that MCID estimates remained consistent (32–34 meters),supporting the robustness of the imputation approach. Missing data occurred in 8% of cases, primarily due to participant dropout, and were addressed using multiple imputation with five iterations to minimize bias.

## Ethical considerations

This study received ethical clearance from the IQRA National University institute review board (application number: 2022/09/CE, on 15th November,2022) and was conducted according to the Declaration of Helsinki. Informed consent was obtained from all participants.

## Results

### Results for 2MWT MCID determination

The following results address the study's aim to determine the MCID of the 2MWT in subacute stroke patients using an anchor-based approach with the ABC_Gait and GPE. The sample included 150 participants from seven Pakistani rehabilitation centers. The participants had a wide range of walking abilities as indicated by the baseline walking distances according to the 2MWT, which ranged from 10 to 150 meters (mean 62.5 meters, SD 38.4 meters). The groups with hemorrhagic stroke showed greater variability in baseline scores (SD = 42.1 meters) compared to those with ischemic stroke (SD = 35.8 meters). The pattern of post-intervention improvement was consistent across all centers, with no significant outliers.

## Participant characteristics

The study involved a total of 150 participants (86 men, 64 women; mean age, 64.2±11.8 years; mean time since stroke onset, 92±46 days). Most participants had an ischemic stroke (68%), and 72% used assistive devices at baseline (Table 2).

## Changes in outcome measures

Post-intervention improvements were observed across all centers, with no statistical outliers (p > 0.05). The 2MWT distance improved by an average of 36.2 meters (p < 0.001), with the mean 2MWT distance rising from 38.4 to 98.7 meters. ABC scores increased by 60.2, compared to 22.1 to 74.8, and ABC_Gait rose by 58.1, compared to 21.4 to 73.5, with a significance level of p = 0.001. GPE scores showed that 80.0% of participants and 76.7% of therapists reported improvement (GPE_pwST and GPE_PT, respectively). The mean (SD) scores on the MBI increased from 72.1 (19.6) to 86.4 (17.3) (p < 0.001).

Fig 2 illustrates the pre- and post-intervention MBI scores, highlighting significant improvements in function. This bar chart shows the mean MBI score increased from 72.1 (SD 19.6) to 86.4 (SD 17.3) (p < 0.001), reflecting enhanced functional independence across daily activities, complementing the mobility gains observed in the 2MWT (Table 3).

## MCID determination

Logistic regression showed a positive association between increased 2MWT and ABC_Gait change scores (OR = 1.15, 95% CI: 1.09–1.21, p < 0.001). A post-hoc subgroup analysis compared 2MWT improvements between centers with (n = 45) and without (n = 105) robotic gait training. The robotic group showed a mean improvement of 37.5±33.2 meters (95% CI: 29.8–45.2) vs. 35.8±32.6 meters (95% CI: 29.4–42.2) for the manual group (p = 0.21, Cohen's d = 0.05), suggesting the 33 meters MCID is robust across intervention modalities.

Stratified analyses demonstrated consistent MCID estimates across subgroups (Table 4), with a 35-meter MCID for participants aged <65 years (n = 78, 95% CI: 31–39) vs. 31 meters for those ≥65 years (n = 72, 95% CI: 28–34, p = 0.15), and 34 meters for ischemic stroke (n = 102, 95% CI: 30–38) vs. 32 meters for hemorrhagic stroke (n = 48, 95% CI: 29–35, p = 0.32). The anchor-based method used for the ABC_Gait subscale (indicating an improvement of 10% or more) produced an MCID of 33 meters (95% CI: 30–36 meters) and an AUC of 0.89 (95% CI: 0.84 to 0.94). The sensitivity was 87%, and the specificity was 82%. Logistic regression showed an association between 2MWT change scores and ABC_Gait change scores (OR = 1.15, 95% CI: 1.09–1.21, p < 0.001).

Fig 3 displays the distribution of 2MWT Improvements by Age Group and Stroke Type. This box plot illustrates the distribution of 2MWT improvements (meters) across age groups (<65 years, ≥65 years) and stroke types (ischemic, hemorrhagic), highlighting median, interquartile ranges, and variability in improvements among subacute stroke patients (n = 150). Fig 4 displays the Receiver Operating Characteristic (ROC) curve for the 2MWT MCID (33 meters), with an area

**Table 2. Baseline characteristics.**

| Characteristic | Value |
|---|---|
| Age (years, mean±SD) | 64.2±11.8 |
| Sex (Male/Female, n) | 86/64 |
| Time from stroke onset (days, mean±SD) | 92±46 |
| Stroke type (Ischemic/Hemorrhagic, %) | 68%/32% |
| Baseline 2MWT (meters, mean±SD) | 62.5±38.4 |
| Baseline MBI (mean±SD) | 72.1±19.6 |
| Assistive device use (Yes/No, %) | 72%/28% |

Table 2 summarizes the baseline demographic and clinical characteristics of the participants.

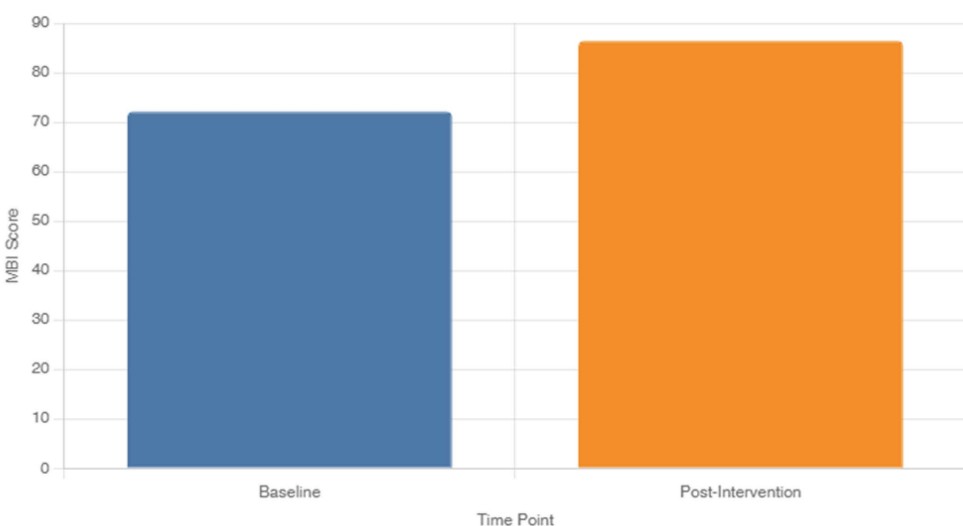

**Fig 2. Pre-and post-intervention MBI scores.**

**Table 3. Pre-post outcomes.**

| Outcome Measure | Baseline (Mean±SD) | Post-Intervention (Mean±SD) | Change (Mean±SD) | p-value |
|---|---|---|---|---|
| 2MWT (meters) | 62.5±38.4 | 98.7±42.1 | 36.2±32.8 | <0.001 |
| ABC (score) | 60.2±22.1 | 74.8±19.3 | 14.6±10.5 | <0.001 |
| ABC_Gait (score) | 58.1±21.4 | 73.5±18.2 | 15.4±11.0 | <0.001 |
| MBI (score) | 72.1±19.6 | 86.4±17.3 | 14.3±9.2 | <0.001 |
| GPE_pwST (% improved) | – | 80.0% | – | – |
| GPE_PT (% improved) | – | 76.7% | – | – |

Table 3 summarizes both the pre intervention and post intervention study outcomes.

**Table 4. Subgroup MCID.**

| Subgroup | N | MCID (meters, 95% CI) | p-value |
|---|---|---|---|
| Age<65 years | 78 | 35 (1–39) | 0.15 |
| Age≥65 years | 72 | 31 (28–34) | 0.15 |
| Ischemic Stroke | 102 | 34 (30–38) | 0.32 |
| Hemorrhagic Stroke | 48 | 32 (29–35) | 0.32 |
| Robotic Gait Training | 45 | 37.5 (29.8–45.2) | 0.21 |
| Manual Gait Training | 105 | 35.8 (29.4–42.2) | 0.21 |

Table 4 summarizes the MCID Determination across the different subgroups.

under the curve (AUC) of 0.89 (95% CI: 0.84–0.94), demonstrating a high sensitivity (87%) and specificity (82%) of the 2MWT MCID estimate and lastly, Fig 5 is a scatter plot illustrating the correlation (r=0.68) between 2MWT change scores (mean 36.2±32.8 meters) and ABC_Gait change scores (mean 15.4±11.0).

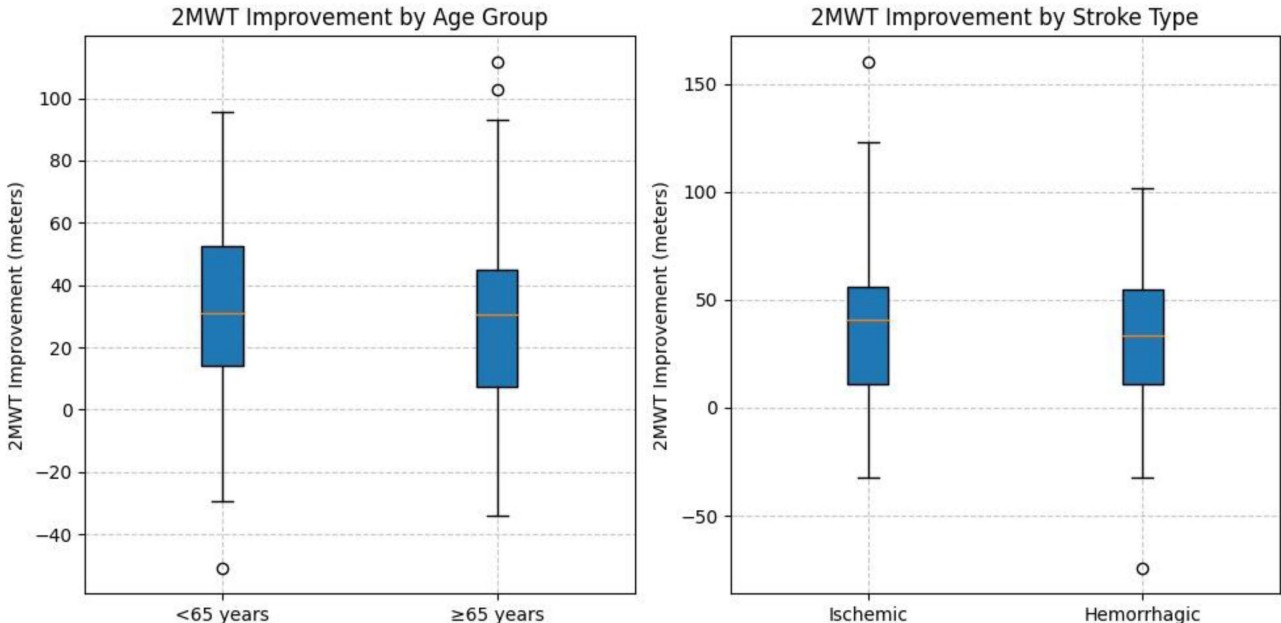

**Fig 3. Distribution of 2MWT Improvements by Age Group and Stroke Type.**

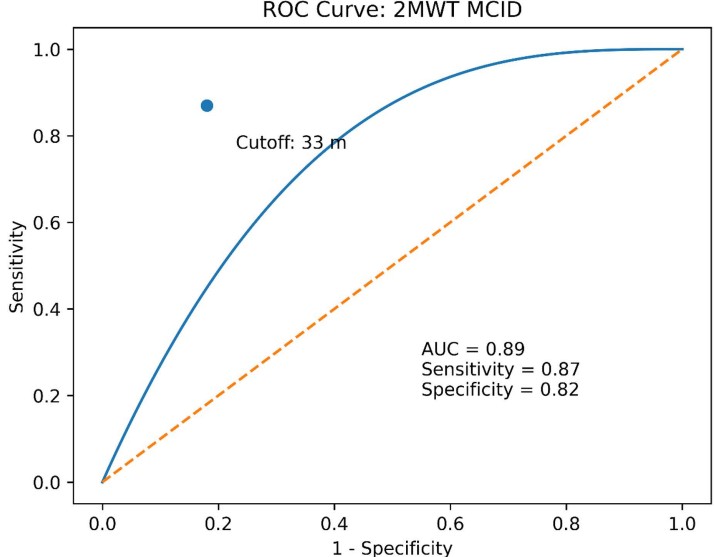

**Fig 4. Receiver Operating Characteristic (ROC) curve for the 2MWT MCID.**

## Discussion

This study establishes an MCID of 33 meters at the 2MWT for adults with subacute strokes, providing a strong baseline for evaluating clinically meaningful changes in walking ability. The larger group (n = 150) offers greater generalizability

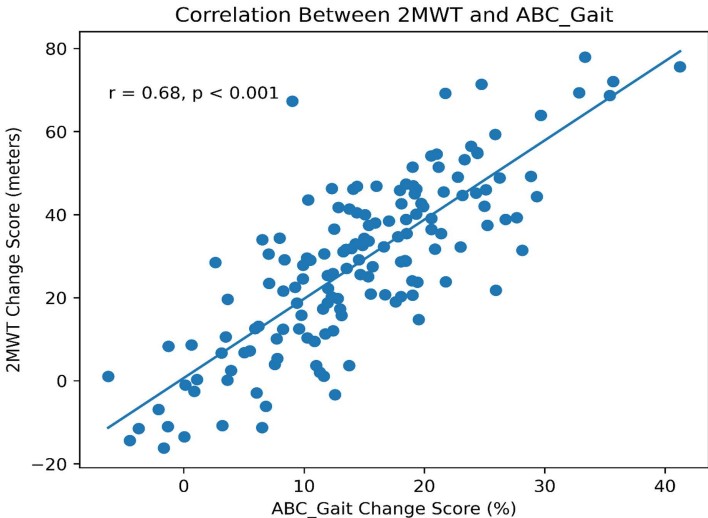

**Fig 5. Correlation (r = 0.68) between 2MWT and ABC_Gait change scores.**

compared to smaller studies [28]. The MCID indicates substantial improvements in gait-specific balance confidence (ABC Gait) and functional independence (MBI), implying that 33 meters increase in 2MWT distance signifies genuine functional progress related to daily mobility. A moderate correlation (r = 0.68, p < 0.001) was observed between the 2MWT and the ABC Gait change scores, reflecting the link between objective walking ability and subjective satisfaction, consistent with recent research on patient-centered outcomes [13,27].

The 33 meters MCID not only quantifies meaningful change but also aligns with practical constraints of clinical settings, as evidenced by its comparison to the 6MWT [29].This practicality enhances its potential as a standard tool, particularly in resource-limited environments. Our estimate not only differs from the 6MWT MCID of 50 meters reported by [27] but also likely results from differences in test duration and anchor measures [8]. Compared to the 2MWT MCID in COPD (26–30 meters [10]), the 33 meters MCID in subacute stroke is higher, likely due to the greater potential for motor recovery in the subacute phase driven by neuroplasticity [3]. In multiple sclerosis, the 2MWT MCID is lower (20–25 meters [26]), possibly reflecting less pronounced gait variability compared to stroke (**Table 5**). These differences underscore the importance of context-specific MCIDs, as stroke-related impairments (e.g., hemiparesis, spasticity) may require larger improvements

**Table 5. Comparison with other populations.**

| Test | Population | MCID (meters) | Reference |
|---|---|---|---|
| 2MWT | Subacute Stroke | 33 | Current Study |
| 2MWT | COPD | 26–30 | [10] |
| 2MWT | Multiple Sclerosis | 20–25 | [26] |
| 10-Meter Walk Test | Stroke | 0.16–0.30 m/s | [27] |

Table 5 summarizes the MCIDs for short-duration walking tests across various populations, providing context for the present study's aims.

to achieve clinical significance. Future research should explore whether these variations are attributable to differences in patient populations, anchor measures, or rehabilitation intensity.

Logistic regression strengthened our MCID by adjusting for baseline and demographic variability. The short duration of the 2MWT, which requires minimal equipment and time, improves its practicality in resource-limited clinical settings. However, future studies should standardize protocols or quantify the effect of specific interventions (e.g., robotic vs. manual gait training) to validate the consistency of the MCID further. Its affordability further supports its suitability as a standard assessment tool [30]. Other reasons related to the length of 33 meters MCID might include differences in rehabilitation levels, patient motivation, or unmeasured characteristics such as spasticity, which could influence perceived changes in walking abilities. For example, spasticity may limit gait efficiency, while motivation could affect engagement in therapy [4]. Future studies should incorporate clinical scales (e.g., Modified Ashworth Scale for spasticity) and patient-reported motivation measures to quantify their impact on MCID estimates. 33 meters increase in the 2MWT indicates meaningful mobility gains thus enabling clinicians to prioritize intensive, tailored interventions to enhance independence in subacute stroke patients.

This study addresses prior limitations, such as small sample sizes, by recruiting 150 participants and validating the ABC_Gait subscale [1,25]. Reliance on self-reported ABC_Gait may introduce recall bias [31], and cultural factors, such as limited access to rehabilitation in rural Pakistan, may affect the generalizability of the findings. Within Pakistan, urban centers often provide better access to rehabilitation facilities and technologies (e.g., robotic gait training), while rural areas face barriers like transportation and limited therapist availability. This disparity may affect the applicability of the 33 meters MCID, as rural patients may have less frequent therapy sessions. Future studies should compare urban and rural cohorts to assess the consistency of the MCID across different socioeconomic contexts. Additionally, environmental factors, such as walkway surface consistency, were controlled by using a standardized 30-meter track, but unmeasured variables like ambient temperature may have influenced 2MWT performance. Assessor variability was minimized through standardized training although minor inter-rater differences (estimated Cohen's $\kappa = 0.85$) may have occurred. and lastly, unmeasured variables, such as spasticity or patient motivation, may influence the 33 meters MCID.

Furthermore, the other limitation was the variability of interventions across centers, such as the use of robotic gait training in some facilities. However, subgroup analyses revealed no significant difference in outcomes between robotic and manual training ($p = 0.21$), suggesting center-specific resources did not bias the MCID. Future studies should prioritize the use of multicenter data with standardized goals to help strengthen the generalizability of our 33 meters MCID benchmark across diverse clinical environments.

## Conclusion

The 33 meters 2MWT MCID provides a reliable, practical benchmark for assessing mobility gains in subacute stroke. Its simplicity supports use in resource-limited settings. However, validation in chronic stroke and exploration of emerging interventions, such as virtual reality(VR) are needed to extend its applicability. Future studies should incorporate wearable sensors to quantify gait dynamics (e.g., stride length, velocity) in real-world settings, thereby further validating the 33 meters MCID.

## Supporting information

**S1 Checklist. Inclusivity in global research questionnaire.**
(DOCX)

**S1 File. Checklist: strobe checklist.**
(DOCX)

## Author contributions

**Conceptualization:** Muslim Khan, Hafiz Shehzad Muzamil, Ahmad M Osailan, Ayman Abdullah Alhammad, Samiullah Khan, Mazhar Mushtaq, Engy BadrEldin Saleh Moustafa, Theeb Naif S Alsalem.

**Data curation:** Hafiz Shehzad Muzamil, Ahmad M Osailan, Ayman Abdullah Alhammad, Samiullah Khan, Mazhar Mushtaq, Engy BadrEldin Saleh Moustafa, Theeb Naif S Alsalem.

**Formal analysis:** Muslim Khan, Hafiz Shehzad Muzamil, Ahmad M Osailan, Ayman Abdullah Alhammad, Samiullah Khan, Mazhar Mushtaq, Engy BadrEldin Saleh Moustafa, Theeb Naif S Alsalem.

**Funding acquisition:** Ahmad M Osailan, Ayman Abdullah Alhammad, Engy BadrEldin Saleh Moustafa, Theeb Naif S Alsalem.

**Investigation:** Muslim Khan, Hafiz Shehzad Muzamil, Ahmad M Osailan, Ayman Abdullah Alhammad, Samiullah Khan, Mazhar Mushtaq, Engy BadrEldin Saleh Moustafa, Theeb Naif S Alsalem.

**Methodology:** Muslim Khan, Hafiz Shehzad Muzamil, Ahmad M Osailan, Ayman Abdullah Alhammad, Samiullah Khan, Mazhar Mushtaq, Engy BadrEldin Saleh Moustafa, Theeb Naif S Alsalem.

**Project administration:** Muslim Khan, Hafiz Shehzad Muzamil, Ahmad M Osailan, Ayman Abdullah Alhammad, Samiullah Khan, Mazhar Mushtaq, Theeb Naif S Alsalem.

**Resources:** Ahmad M Osailan, Samiullah Khan, Mazhar Mushtaq, Engy BadrEldin Saleh Moustafa, Theeb Naif S Alsalem.

**Software:** Muslim Khan, Samiullah Khan, Theeb Naif S Alsalem.

**Supervision:** Muslim Khan, Hafiz Shehzad Muzamil, Ahmad M Osailan, Samiullah Khan, Engy BadrEldin Saleh Moustafa.

**Validation:** Edward Muteesasira, Hafiz Shehzad Muzamil, Ahmad M Osailan, Ayman Abdullah Alhammad, Samiullah Khan, Engy BadrEldin Saleh Moustafa, Theeb Naif S Alsalem.

**Visualization:** Edward Muteesasira, Muslim Khan, Hafiz Shehzad Muzamil, Ahmad M Osailan, Ayman Abdullah Alhammad, Samiullah Khan, Mazhar Mushtaq, Engy BadrEldin Saleh Moustafa, Theeb Naif S Alsalem.

**Writing – original draft:** Muslim Khan, Hafiz Shehzad Muzamil, Ahmad M Osailan, Ayman Abdullah Alhammad, Samiullah Khan, Mazhar Mushtaq, Engy BadrEldin Saleh Moustafa, Theeb Naif S Alsalem.

**Writing – review & editing:** Muslim Khan, Hafiz Shehzad Muzamil, Ahmad M Osailan, Ayman Abdullah Alhammad, Samiullah Khan, Mazhar Mushtaq, Engy BadrEldin Saleh Moustafa, Theeb Naif S Alsalem.

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
