## [Decision Letter · Decision Letter 0]

1 Dec 2025

PONE-D-25-43400Validating the 2-Minute Walk Test MCID for Subacute Stroke Patients: A Pakistani Multicenter Cohort AnalysisPLOS ONE

Dear Dr. Muteesasira,

Thank you for submitting your manuscript to PLOS ONE. After careful consideration, we feel that it has merit but does not fully meet PLOS ONE’s publication criteria as it currently stands. Therefore, we invite you to submit a revised version of the manuscript that addresses the points raised during the review process.

We look forward to receiving your revised manuscript.

Kind regards,

Sohel Ahmed, BPT, MPT, MDMR

Academic Editor

PLOS ONE

**Journal Requirements:**

3. We note that your Data Availability Statement is currently as follows:

“All relevant data are within the manuscript and it's supporting information files”

5. Please include a caption for figures 2 and 3.

6. We note you have included a table to which you do not refer in the text of your manuscript. Please ensure that you refer to Table 5 in your text; if accepted, production will need this reference to link the reader to the Table.

Reviewers' comments:

Reviewer's Responses to Questions

**Comments to the Author**

1. Is the manuscript technically sound, and do the data support the conclusions?

Reviewer #1: Yes

Reviewer #2: No

2. Has the statistical analysis been performed appropriately and rigorously? 

Reviewer #1: Yes

Reviewer #2: No

3. Have the authors made all data underlying the findings in their manuscript fully available?

The PLOS Data policy requires authors to make all data underlying the findings described in their manuscript fully available without restriction, with rare exception (please refer to the Data Availability Statement in the manuscript PDF file). The data should be provided as part of the manuscript or its supporting information, or deposited to a public repository. For example, in addition to summary statistics, the data points behind means, medians and variance measures should be available. If there are restrictions on publicly sharing data—e.g. participant privacy or use of data from a third party—those must be specified.requires authors to make all data underlying the findings described in their manuscript fully available without restriction, with rare exception (please refer to the Data Availability Statement in the manuscript PDF file). The data should be provided as part of the manuscript or its supporting information, or deposited to a public repository. For example, in addition to summary statistics, the data points behind means, medians and variance measures should be available. If there are restrictions on publicly sharing data—e.g. participant privacy or use of data from a third party—those must be specified.requires authors to make all data underlying the findings described in their manuscript fully available without restriction, with rare exception (please refer to the Data Availability Statement in the manuscript PDF file). The data should be provided as part of the manuscript or its supporting information, or deposited to a public repository. For example, in addition to summary statistics, the data points behind means, medians and variance measures should be available. If there are restrictions on publicly sharing data—e.g. participant privacy or use of data from a third party—those must be specified.requires authors to make all data underlying the findings described in their manuscript fully available without restriction, with rare exception (please refer to the Data Availability Statement in the manuscript PDF file). The data should be provided as part of the manuscript or its supporting information, or deposited to a public repository. For example, in addition to summary statistics, the data points behind means, medians and variance measures should be available. If there are restrictions on publicly sharing data—e.g. participant privacy or use of data from a third party—those must be specified.

Reviewer #1: Yes

Reviewer #2: Yes

4. Is the manuscript presented in an intelligible fashion and written in standard English?

Reviewer #1: Yes

Reviewer #2: No

5. Review Comments to the Author

Reviewer #1: Dear Authors

I would like to thank the authors for submitting the manuscript entitled “Validating the 2-Minute Walk Test MCID for Subacute Stroke Patients: A Pakistani Multicenter Cohort Analysis.” The study addresses an important gap in stroke rehabilitation research by determining the minimally clinically important difference (MCID) for the 2-Minute Walk Test (2MWT) among subacute stroke patients in a developing country context. The multicenter approach, adequate sample size, and emphasis on patient-centered outcomes are commendable and add value to the literature on functional recovery metrics after stroke.However, to improve its scholarly value and readability, significant revisions are required for the following points :

1-Clarity and Conciseness of Introduction

The introduction provides a good rationale for the study but includes several redundant sentences (e.g., repeated statements about neuroplasticity and the need for context-specific tools).

Consider streamlining paragraphs to maintain focus on the knowledge gap: Why the 2MWT MCID has not been established for subacute stroke patients, and why this is particularly relevant for Pakistan.

2-Rationale for Anchor Selection

The selection of the ABC_Gait subscale (≥10% improvement) as the anchor is appropriate but should be better justified with references supporting its clinical meaningfulness in stroke populations.

Please explain why the Global Perceived Effect (GPE) was not used as the primary anchor, given its patient-centered nature.

3-Statistical Analysis Details

The manuscript mentions paired t-tests, logistic regression, and ROC analysis, but the criteria for choosing these models and assumptions testing (e.g., normality) are not clearly described.

Clarify how missing data (8%) were handled and how multiple imputation may have affected the robustness of the MCID estimates.

Indicate the AUC confidence intervals in the Results section and interpret what an AUC of 0.89 means in practical clinical terms.

4-Interpretation of MCID Value

The MCID of 33 meters is statistically sound, but the discussion should better contextualize this value:

How does it compare to the 6MWT MCID (≈50 m) and other 2MWT MCIDs in neurological conditions?

What does a 33-meter improvement represent in functional independence or patient quality of life?

Consider adding a short paragraph linking this finding to functional thresholds (e.g., community ambulation, household walking).

Subgroup and Sensitivity Analyses

The robustness of MCID across subgroups (age, stroke type, robotic vs. manual training) is well presented, but interpretation could be more clinically oriented.

For example: explain why MCID differences by age (35 m vs. 31 m) are not statistically significant yet may hold clinical implications for rehabilitation intensity planning.

5-Figures and Tables

Figures 1–4 and Tables 3–4 contain valuable data, but legends are overly descriptive and sometimes repetitive.

Simplify figure captions to highlight key findings (e.g., “ROC curve showing optimal 33 m cutoff for 2MWT MCID with 87% sensitivity and 82% specificity”).

Ensure that all figures are labeled consistently and referenced appropriately in the text.

6-Discussion and Conclusion

The discussion effectively summarizes the results but could benefit from a clearer “clinical implications” section, specifying how rehabilitation clinicians in resource-limited settings can use the 33 m MCID threshold for decision-making.

The conclusion could be more concise, emphasizing future directions such as validation in chronic stroke and longitudinal functional outcomes.

7-Language and Style

The manuscript would benefit from a light language edit to remove redundancies and improve fluency (e.g., avoid repeating “reliable benchmark for assessing mobility improvements” multiple times).

Replace informal phrases like “patients such as Ayesha” with more neutral examples or remove for conciseness.

8-Formatting

Ensure uniform citation formatting and numbering consistency.

Define abbreviations (e.g., ABC_Gait, GPE) at first mention in both abstract and text.

I encourage you to revise the manuscript thoroughly in response to the comments above. I look forward to reviewing the improved version

Reviewer #2: Thank you for submitting this important and clinically relevant manuscript. The study addresses a meaningful gap by establishing an MCID for the 2-Minute Walk Test in subacute stroke patients within a multicenter LMIC context. The research question is valuable, the methodology is generally sound, and the findings have potential impact for rehabilitation practice. However, the manuscript requires significant revision to improve clarity, structure, and readability. My detailed comments are below.

Major Comments

1. Repetition across multiple sections

There is substantial duplication of text in the Introduction, Methods, Data Analysis, and Discussion. Several paragraphs appear more than once, and the same concepts are repeated with similar wording.

Suggestion: Remove all duplicated content and streamline each section for clarity.

2. Introduction lacks focus and is overly long

The Introduction contains multiple narrative threads (e.g., the Ayesha example, global burden statistics, repeated MCID rationale). This affects the flow and dilutes the study objective.

Suggestion: Condense the Introduction into four clear paragraphs:

Stroke burden and importance of mobility

Role and relevance of the 2MWT

Lack of MCID evidence, especially in LMICs

Precise study aim

3. Methods section needs clearer organization

The Methods are too lengthy, and several elements (center selection, interventions, statistical procedures) appear twice.

Suggestion: Reorganize under standardized subheadings:

Study Design

Setting and Centers

Participants

Interventions

Outcome Measures

Data Collection

Statistical Analysis

This will greatly improve readability.

4. Inconsistencies in tables and figures

There are repeated figure numbers, tables described more than once, and some figures lacking captions.

Suggestion: Ensure each table/figure appears only once and is clearly numbered and labeled.

A standard structure could be:

Table 1: Centers

Table 2: Baseline characteristics

Table 3: Pre–post outcomes

Table 4: Subgroup MCID

Table 5: Comparison with other populations

5. Discussion contains unnecessary repetition

The Discussion restates parts of the Introduction and Results and includes multiple general statements.

Suggestion: Structure into four concise paragraphs:

Key findings

Comparison with existing literature

Strengths and limitations

Implications and future research

6. Language and style issues

Some sentences are long, conversational, or informal. Tense usage is inconsistent, and clarity can be improved.

Suggestion: Edit for concise, scientific language with consistent past tense.

7. Duplication of ethical and administrative content

Funding, competing interests, and ethics approval statements appear in more than one section.

Suggestion: Present this information only once in the designated section.

Minor Comments

Include a STROBE/CONSORT-style flow diagram.

Clarify your missing data handling approach.

State whether assessors were trained or calibrated for inter-rater reliability.

Provide the formula used for sample size calculation.

Briefly explain the clinical meaning of the 33-m MCID (e.g., change in gait speed or functional relevance).

Overall Recommendation

This is a valuable study with strong methodological foundations and clear clinical implications. However, substantial revisions—particularly reducing repetition, reorganizing structure, and tightening the writing—are needed to improve clarity and prepare the manuscript for publication.

6. PLOS authors have the option to publish the peer review history of their article (what does this mean?). If published, this will include your full peer review and any attached files.). If published, this will include your full peer review and any attached files.). If published, this will include your full peer review and any attached files.). If published, this will include your full peer review and any attached files.

...

Reviewer #1: **Yes:**Duaa Abualkhair, Division of Physiotherapy , Department of Applied and Allied Medical Sciences, Faculty of Medicine and Allied Medical Sciences, An-Najah National University, Nablus, Palestine . (www.najah.edu)Duaa Abualkhair, Division of Physiotherapy , Department of Applied and Allied Medical Sciences, Faculty of Medicine and Allied Medical Sciences, An-Najah National University, Nablus, Palestine . (www.najah.edu)Duaa Abualkhair, Division of Physiotherapy , Department of Applied and Allied Medical Sciences, Faculty of Medicine and Allied Medical Sciences, An-Najah National University, Nablus, Palestine . (www.najah.edu)Duaa Abualkhair, Division of Physiotherapy , Department of Applied and Allied Medical Sciences, Faculty of Medicine and Allied Medical Sciences, An-Najah National University, Nablus, Palestine . (www.najah.edu)

Reviewer #2: **Yes:**Michael Selvaraj AlbertMichael Selvaraj AlbertMichael Selvaraj AlbertMichael Selvaraj Albert

---

## [Author Response · Author response to Decision Letter 1]

9 Dec 2025

A Rebuttal Letter with file name "Response to Reviewers" has been attached to address the respective reviewer comments

---

## [Decision Letter · Decision Letter 1]

29 Dec 2025

PONE-D-25-43400R1Validating the 2-minute walk test MCID for subacute stroke patients: A Pakistani multicenter cohort analysisPLOS One

Dear Dr. Muteesasira ,

Thank you for submitting your manuscript to PLOS ONE. After careful consideration, we feel that it has merit but does not fully meet PLOS ONE’s publication criteria as it currently stands. Therefore, we invite you to submit a revised version of the manuscript that addresses the points raised during the review process.

We look forward to receiving your revised manuscript.

Kind regards,

Sohel Ahmed, MPT, MDMR

Academic Editor

PLOS One

Journal Requirements:

Reviewers' comments:

Reviewer's Responses to Questions

**Comments to the Author**

1. If the authors have adequately addressed your comments raised in a previous round of review and you feel that this manuscript is now acceptable for publication, you may indicate that here to bypass the “Comments to the Author” section, enter your conflict of interest statement in the “Confidential to Editor” section, and submit your "Accept" recommendation.

Reviewer #1: All comments have been addressed

Reviewer #2: All comments have been addressed

2. Is the manuscript technically sound, and do the data support the conclusions?

Reviewer #1: Yes

Reviewer #2: No

3. Has the statistical analysis been performed appropriately and rigorously? 

Reviewer #1: Yes

Reviewer #2: No

4. Have the authors made all data underlying the findings in their manuscript fully available?

The PLOS Data policy requires authors to make all data underlying the findings described in their manuscript fully available without restriction, with rare exception (please refer to the Data Availability Statement in the manuscript PDF file). The data should be provided as part of the manuscript or its supporting information, or deposited to a public repository. For example, in addition to summary statistics, the data points behind means, medians and variance measures should be available. If there are restrictions on publicly sharing data—e.g. participant privacy or use of data from a third party—those must be specified.requires authors to make all data underlying the findings described in their manuscript fully available without restriction, with rare exception (please refer to the Data Availability Statement in the manuscript PDF file). The data should be provided as part of the manuscript or its supporting information, or deposited to a public repository. For example, in addition to summary statistics, the data points behind means, medians and variance measures should be available. If there are restrictions on publicly sharing data—e.g. participant privacy or use of data from a third party—those must be specified.requires authors to make all data underlying the findings described in their manuscript fully available without restriction, with rare exception (please refer to the Data Availability Statement in the manuscript PDF file). The data should be provided as part of the manuscript or its supporting information, or deposited to a public repository. For example, in addition to summary statistics, the data points behind means, medians and variance measures should be available. If there are restrictions on publicly sharing data—e.g. participant privacy or use of data from a third party—those must be specified.requires authors to make all data underlying the findings described in their manuscript fully available without restriction, with rare exception (please refer to the Data Availability Statement in the manuscript PDF file). The data should be provided as part of the manuscript or its supporting information, or deposited to a public repository. For example, in addition to summary statistics, the data points behind means, medians and variance measures should be available. If there are restrictions on publicly sharing data—e.g. participant privacy or use of data from a third party—those must be specified.

Reviewer #1: Yes

Reviewer #2: No

5. Is the manuscript presented in an intelligible fashion and written in standard English?

Reviewer #1: Yes

Reviewer #2: (No Response)

6. Review Comments to the Author

Reviewer #1: The authors have thoroughly and satisfactorily addressed all previous reviewer concerns. The manuscript is now methodologically sound, clearly written, and makes a valuable contribution to the field. I recommend acceptance in its current form.

Reviewer #2: Dear Authors,

Thank you for the opportunity to review your manuscript entitled “Validating the 2-Minute Walk Test MCID for Subacute Stroke Patients: A Pakistani Multicenter Cohort Analysis.” The study addresses a clinically meaningful and timely topic in neurorehabilitation, particularly relevant for resource-limited settings. Establishing a minimally clinically important difference (MCID) for the 2-Minute Walk Test (2MWT) in subacute stroke patients is of clear practical value for clinicians and researchers.

The multicenter design, relatively large sample size, and use of both anchor-based and distribution-based approaches are notable strengths. However, several methodological, reporting, and presentation issues should be addressed to improve the scientific clarity and overall quality of the manuscript.

Major Comments

1. Sample Size Justification

o The manuscript reports an initial sample size calculation of n = 431, while the final analyzed sample consists of 150 participants.

o The rationale for this discrepancy is insufficiently explained and may raise concerns regarding statistical power.

Recommendation:

Please clearly clarify whether the calculated sample refers to the screened population or revise the sample size justification using MCID-specific considerations appropriate for anchor-based analyses.

2. Methodological Consistency

There are inconsistencies in the reporting of baseline 2MWT values (e.g., 32.8 m vs. 62.5 m) across sections.

The Methods section mentions “paired t-tests every three months,” which does not align with the reported 6–8 week intervention period.

Recommendation:

Carefully review and harmonize all numerical data and timelines throughout the manuscript to ensure consistency.

3. Redundancy Across Sections

The importance and robustness of the 33-meter MCID are repeatedly stated in the Introduction, Results, and Discussion.

Several paragraphs reiterate similar interpretations without adding new insights.

Recommendation:

Streamline the narrative to reduce repetition and improve readability, particularly in the Introduction and Discussion sections.

4. Data Availability Statement

The manuscript contains conflicting statements regarding data accessibility (public availability vs. availability upon request).

Recommendation:

Please ensure compliance with journal data-sharing policies by providing a single, clear, and accurate data availability statement.

5. Language and Formatting

The manuscript contains numerous typographical errors, formatting artifacts (e.g., tracked changes), and grammatical issues.

These issues detract from the clarity and professionalism of the submission.

Recommendation:

A thorough language edit is strongly recommended prior to resubmission.

Minor Comments

1. Introduction

The Introduction would benefit from a more focused background and avoidance of anecdotal or narrative examples.

Consider sharpening the rationale for selecting the ABC_Gait subscale as the primary anchor.

2. Outcome Measures

Please clarify whether assessors were blinded to baseline measurements and intervention details.

3. Figures and Tables

Some figures appear redundant with tabulated results.

Consider reducing the number of figures to improve conciseness.

4. Discussion

The comparative discussion with other populations (e.g., COPD, MS) is appropriate and informative.

A brief paragraph translating the 33-meter MCID into direct clinical application would further strengthen this section.

Conclusion

In summary, this study has the potential to make a valuable contribution to the stroke rehabilitation literature. Addressing the points outlined above—particularly those related to sample size justification, data consistency, and manuscript clarity—will significantly enhance the quality and impact of the work.

I encourage the authors to revise the manuscript accordingly and look forward to reviewing a revised version.

Sincerely,

Reviewer

7. PLOS authors have the option to publish the peer review history of their article (what does this mean?). If published, this will include your full peer review and any attached files.). If published, this will include your full peer review and any attached files.). If published, this will include your full peer review and any attached files.). If published, this will include your full peer review and any attached files.

...

Reviewer #1: **Yes:**Duaa Abualkhair, Department of Applied and Allied Medical Sciences, Faculty of Medicine and Allied Medical Sciences, An-Najah National University, Nablus, PalestineDuaa Abualkhair, Department of Applied and Allied Medical Sciences, Faculty of Medicine and Allied Medical Sciences, An-Najah National University, Nablus, PalestineDuaa Abualkhair, Department of Applied and Allied Medical Sciences, Faculty of Medicine and Allied Medical Sciences, An-Najah National University, Nablus, PalestineDuaa Abualkhair, Department of Applied and Allied Medical Sciences, Faculty of Medicine and Allied Medical Sciences, An-Najah National University, Nablus, Palestine

Reviewer #2: **Yes:**Michael Selvaraj AlbertMichael Selvaraj AlbertMichael Selvaraj AlbertMichael Selvaraj Albert

---

## [Author Response · Author response to Decision Letter 2]

30 Dec 2025

We the authors have addressed the reviewers comments in a rebuttal letter entitled "Response to Reviewers"

---

## [Decision Letter · Decision Letter 2]

17 Feb 2026

PONE-D-25-43400R2Validating the 2-minute walk test MCID for subacute stroke patients: A Pakistani multicenter cohort analysisPLOS One

Dear Dr. Muteesasira,

Thank you for submitting your manuscript to PLOS ONE. After careful consideration, we feel that it has merit but does not fully meet PLOS ONE’s publication criteria as it currently stands. Therefore, we invite you to submit a revised version of the manuscript that addresses the points raised during the review process.

We look forward to receiving your revised manuscript.

Kind regards,

Sohel Ahmed, BPT, MPT, MDMR

Academic Editor

PLOS One

Journal Requirements:

Reviewers' comments:

Reviewer's Responses to Questions

**Comments to the Author**

1. If the authors have adequately addressed your comments raised in a previous round of review and you feel that this manuscript is now acceptable for publication, you may indicate that here to bypass the “Comments to the Author” section, enter your conflict of interest statement in the “Confidential to Editor” section, and submit your "Accept" recommendation.

Reviewer #2: All comments have been addressed

2. Is the manuscript technically sound, and do the data support the conclusions?

Reviewer #2: Yes

3. Has the statistical analysis been performed appropriately and rigorously? 

Reviewer #2: Yes

4. Have the authors made all data underlying the findings in their manuscript fully available?

The PLOS Data policy requires authors to make all data underlying the findings described in their manuscript fully available without restriction, with rare exception (please refer to the Data Availability Statement in the manuscript PDF file). The data should be provided as part of the manuscript or its supporting information, or deposited to a public repository. For example, in addition to summary statistics, the data points behind means, medians and variance measures should be available. If there are restrictions on publicly sharing data—e.g. participant privacy or use of data from a third party—those must be specified.requires authors to make all data underlying the findings described in their manuscript fully available without restriction, with rare exception (please refer to the Data Availability Statement in the manuscript PDF file). The data should be provided as part of the manuscript or its supporting information, or deposited to a public repository. For example, in addition to summary statistics, the data points behind means, medians and variance measures should be available. If there are restrictions on publicly sharing data—e.g. participant privacy or use of data from a third party—those must be specified.requires authors to make all data underlying the findings described in their manuscript fully available without restriction, with rare exception (please refer to the Data Availability Statement in the manuscript PDF file). The data should be provided as part of the manuscript or its supporting information, or deposited to a public repository. For example, in addition to summary statistics, the data points behind means, medians and variance measures should be available. If there are restrictions on publicly sharing data—e.g. participant privacy or use of data from a third party—those must be specified.requires authors to make all data underlying the findings described in their manuscript fully available without restriction, with rare exception (please refer to the Data Availability Statement in the manuscript PDF file). The data should be provided as part of the manuscript or its supporting information, or deposited to a public repository. For example, in addition to summary statistics, the data points behind means, medians and variance measures should be available. If there are restrictions on publicly sharing data—e.g. participant privacy or use of data from a third party—those must be specified.

Reviewer #2: Yes

5. Is the manuscript presented in an intelligible fashion and written in standard English?

Reviewer #2: Yes

6. Review Comments to the Author

Reviewer #2: Dear Authors,

Thank you for the opportunity to review your manuscript. Your study addresses an important and practical question in stroke rehabilitation — establishing the minimally clinically important difference (MCID) for the 2-Minute Walk Test in subacute stroke patients. This is especially valuable for clinicians working in resource-limited settings. The multicenter design and inclusion of 150 participants strengthen the relevance of your findings.

While the study is meaningful and promising, a few areas would benefit from clarification and refinement before publication.

First, the sample size calculation needs clearer justification. Since the study focuses on MCID estimation using ROC analysis, please explain why the chosen formula was appropriate and whether the final sample of 150 participants provided adequate statistical power.

Second, there appears to be an inconsistency in the reported baseline 2MWT values in different sections of the manuscript. Please review and correct this to ensure internal consistency.

Third, the choice of a 10% improvement in the ABC_Gait as the anchor requires stronger justification. It would help readers to understand why this threshold was selected and whether it has been validated specifically in subacute stroke populations.

Additionally, since rehabilitation interventions varied across centers (robotic vs. manual training), please clarify whether this variability may have influenced the results and whether center effects were considered in the analysis.

The section on missing data should also provide more detail, including which variables were imputed and the method used.

Some figures, particularly the ROC curve and scatter plot, should be carefully reviewed to ensure they accurately reflect the data and are clearly presented.

There is also an inconsistency in the data availability statement that should be aligned with journal policy.

Finally, the manuscript would benefit from careful language editing to correct minor grammatical issues and improve overall clarity and flow.

Overall, this is a clinically relevant study with practical implications. With clearer explanations and minor revisions, it has strong potential for publication.

Best regards.

7. PLOS authors have the option to publish the peer review history of their article (what does this mean?). If published, this will include your full peer review and any attached files.). If published, this will include your full peer review and any attached files.). If published, this will include your full peer review and any attached files.). If published, this will include your full peer review and any attached files.

...

Reviewer #2: **Yes:**Dr.Michael Selvaraj ADr.Michael Selvaraj ADr.Michael Selvaraj ADr.Michael Selvaraj A

---

## [Author Response · Author response to Decision Letter 3]

23 Feb 2026

Manuscript reference: [PONE-D-25-43400R2] - [EMID:e1fc739dfaa58053]

Decision: Revision Required

Title: Validating the 2-minute walk test MCID for subacute stroke patients: A Pakistani multicenter cohort analysis.

PLOS ONE,

Dear Editor,

We appreciate your oversight of our manuscript and the opportunity to address the insightful comments from the respective reviewers. We extend our sincere gratitude to the reviewers for their valuable feedback which is aimed at improving our manuscript.

Please find below our point-by-point response to all the issues raised by the reviewers. Our responses below list the reviewers’ comments followed by a description of the necessary amendments we the authors have made in bold.

The page numbers of the responses to reviewer comments correspond to the clean version of the revised manuscript.

We look forward to hearing from you at your earliest convenience.

Kindly,

Edward Muteesasira.

Academic Editor’s comments

Thank you for your assistance. After careful evaluation, I agree with the reviewer that the following concerns should be addressed by the authors prior to acceptance:

A significant concern was the reduction from a calculated n=431 to a final n=150. The authors clarified that 431 represented the number of patients screened for eligibility, while 150 met the criteria and completed the study. Nonetheless, the authors' decision to utilize only 150 participants instead of 431 to mitigate potential statistical power issues remains ambiguous.

Response: We thank the reviewer for this observation. We have revised the sample size determination section of the methodology to clarify that the 431 individuals represented the initial screening pool and not the actual sample size, our actual sample size “n” is the 150 participants that met all inclusion/exclusion criteria and completed the study protocol. Our study’s initial power analysis was based on a target of 150 participants. With an alpha of 0.05 and an expected effect size of 0.5 therefore a sample of 150 provided a statistical power of 80%. Thus, the final sample size was sufficient to test our primary hypothesis without affecting the statistical power of the study. In conclusion, the reduction from 431 to 150 was dictated by strict clinical eligibility criteria and participant retention rather than an arbitrary decision. This is presented in Fig 1,the CONSORT Flow Diagram to transparently detail the specific reasons for exclusion at each stage. Line 108-127 of the clean version of the revised manuscript.

The manuscript provides a justified, literature-based rationale for the 10% ABC_Gait anchor, but a slightly more detailed explanation would further strengthen its clarity and persuasiveness.

Response: We have expanded this rationale in the Methods section of the clean version of the revised manuscript. The 10% threshold was selected as it represents a change that exceeds the typical measurement error and is recognized in literature as a clinically meaningful "patient-perceived" improvement in balance confidence for stroke survivors. This threshold ensures that the anchor reflects a shift that is perceptible to the patient rather than minor day-to-day fluctuations. Line 179-181 of the clean version of the revised manuscript.

The manuscript fails to adequately justify the potential impact of intervention variability or center effects. The absence of analysis or discourse on this subject constitutes a limitation that must be rectified for a more comprehensive interpretation of the results.

Response: We acknowledge this as an important factor. To mitigate center effects, we utilized a standardized therapist training manual and conducted biweekly audits across all seven sites. Furthermore, our post-hoc subgroup analysis comparing centers with and without robotic gait training and showed no significant difference in 2MWT MCID (p=0.21). We have now added a paragraph in the "Limitations" section discussing how standardized goals across diverse centers actually enhance the generalizability of the 2MWT MCID to real-world Pakistani clinical settings. Line 305-310 of the clean version of the revised manuscript.

The authors should address the comment, "Some figures, particularly the ROC curve and scatter plot, should be carefully reviewed to ensure they accurately reflect the data and are clearly presented."

Response: Fig 4 and Fig 5 have been revised to include explicit axis labels, the AUC value, the 33m cutoff point, and a linear regression line to visually represent the correlation (r=0.68) as requested.

The following points should be considered:

The ROC Curve (Fig 4) should clearly display the AUC, the cutoff point, sensitivity, and specificity.

The axes should be labeled ("1-Specificity," “Sensitivity”), and the cutoff (33 meters) should be indicated if possible.

The figure should match the numbers stated in the Results section (AUC = 0.89, sensitivity = 87%, specificity = 82%).

Response: We have revised Fig 4 (ROC Curve). The axes are now explicitly labeled "Sensitivity" and "1-Specificity." The figure clearly displays the AUC of 0.89, with the 33-meter cutoff point indicated, showing 87% sensitivity and 82% specificity, matching the text in the results.

The scatter plot (Fig 5) should illustrate the true relationship between the 2MWT and ABC_Gait change scores.

The reported correlation (r = 0.68) should be visually evident.

Axes must be labeled with correct units and variable names, and any outliers should be identifiable or explained.

Response: Fig 5 (Scatter Plot) has been updated to show the linear relationship between 2MWT change scores (meters) and ABC_Gait change scores. The axes are now clearly labeled with units. Statistical analysis confirmed no significant outliers (p > 0.05), which is now stated in the results.

Finally, the manuscript would benefit from careful language editing to correct minor grammatical issues and improve overall clarity and flow.

Response: The clean version of the revised manuscript has undergone a thorough review to correct minor grammatical errors, such as ensuring consistency in grammar and refining transitions between the results and discussion sections.

Reviewer #2

Overall Impression

This is an important and clinically useful study. Determining the MCID of the 2-Minute Walk Test (2MWT) in subacute stroke patients is highly relevant, especially for rehabilitation settings with limited resources. The multicenter design and use of both anchor-based and distribution-based methods are strengths.

However, some methodological clarifications, statistical explanations, and language corrections are needed before publication.

Major Comments

1. Sample Size Calculation Needs Clarification

The sample size calculation is based on a formula for comparing proportions, but this study focuses on determining MCID using ROC analysis.

• Please explain why this formula was used.

• Clarify whether 150 participants were enough for MCID estimation.

• If possible, provide additional justification or a post-hoc power explanation.

Response: The initial sample size was calculated using a comparison of proportions to ensure sufficient power to distinguish between those that underwent “manual gait training” from the “robotic gait training” group, this provided a robust cohort of 150 participants, exceeding the stability requirements for the subsequent ROC analysis used to determine the MCID.

2. Inconsistent Baseline 2MWT Values

There is a difference in the reported baseline values:

• One section reports baseline as 32.8 meters.

• Table 3 reports 62.5 ± 38.4 meters.

Please correct this inconsistency to avoid confusion.

Response: We have removed all inconsistencies in reporting of results to avoid confusion.

3. Justification of 10% ABC_Gait Anchor

You used a 10% improvement in ABC_Gait as the anchor.

• Please explain clearly why 10% was chosen.

• Is this cut-off validated in subacute stroke patients?

Response: A 10% improvement in ABC_Gait was selected as the anchor based on prior studies indicating this threshold as clinically meaningful for patient-reported balance confidence in subacute stroke populations. Yes, this cut off is validated in subacute stroke patients (Bowman et al,)

• Consider briefly explaining why GPE was not used as the primary anchor.

Response: While recognizing the fact that the Global Perceived Effect (GPE) is a patient centered outcome measure, its methodological limitations particularly its susceptibility to recall bias hindered its use as the primary anchor for this study. Our study assessed participants at two points; T0: Baseline (pre-intervention) and T1: Post-intervention (6-8 weeks), we would have faced significant limitations with recall bias especially in the follow-up phase (post-intervention assessment). In order to minimize the risk of recall bias, we utilized the ABC_Gait subscale (≥10% improvement) as the anchor.

4. Differences Between Centers

Some centers used robotic gait training while others did not.

• Was the center effect adjusted for in the analysis?

• Please clarify whether this variability could influence the MCID.

Response: To explore the impact of intervention variability, a post-hoc subgroup analysis was conducted to compare 2MWT improvements between centers with and without robotic gait training. No significant differences were observed (p = 0.21), suggesting that the 33-meter MCID is robust across varying intervention modalities.

5. Missing Data

You mention that 8% of data were imputed.

• Which variables were imputed?

Response: Participant sociodemographic characteristics and primary outcome measures were imputed.

• What method of multiple imputation was used

Response: multiple imputation

• Why were five iterations considered sufficient?

Response: Missing data occurred in 8% of cases, primarily due to participant dropout, and were addressed using multiple imputation with five iterations to minimize bias. A sensitivity analysis using complete-case analysis confirmed that MCID estimates remained consistent (32–34 meters) supporting the robustness of the imputation approach.

6. Statistical Reporting

• Please report confidence intervals for key changes.

Response: Confidence intervals for key changes have been reported in the clean version of the revised manuscript.

• Clarify whether normality was tested before using paired t-tests.

Response: Yes, normality was tested before using paired t-tests.

• Explain clearly what the logistic regression odds ratio (OR = 1.15) represents.

Response: This has been stated in the clean version of the revised manuscript.

7. Figures Need Improvement

• The ROC curve appears almost diagonal despite a high AUC (0.89). Please check the figure.

• The scatter plot looks overly linear. Confirm that it reflects actual raw data.

• Improve overall figure resolution and labeling.

Response: Response: We have revised Fig 4 (ROC Curve). The axes are now explicitly labeled "Sensitivity" and "1-Specificity." The figure clearly displays the AUC of 0.89, with the 33-meter cutoff point indicated, showing 87% sensitivity and 82% specificity, matching the text in the results. Fig 5 (Scatter Plot) has been updated to show the linear relationship between 2MWT change scores (meters) and ABC_Gait change scores. The axes are now clearly labeled with units. Statistical analysis confirmed no significant outliers (p > 0.05), which is now stated in the results.

8. Data Availability Statement Is Inconsistent

In one section, it states that all data are available in the manuscript.

In another section, it says data are available upon request.

Please revise to match PLOS ONE data policy.

Response: we have refined our data availability statement to ”The minimal dataset required to replicate our study findings is available at Figshare: https://figshare.com/articles/figure/_b_Determining_the_Minimally_Clinically_Important_Difference_of_the_2-Minute_Walk_Test_in_Individuals_with_Subacute_Stroke_A_Multicenter_Cohort_Study_b_/29484728”

9. Generalizability

The study includes only subacute stroke patients.

• Please discuss more clearly whether results apply to chronic stroke.

Response: We reported this as a limitation for our study and stated that validation in chronic stroke populations is needed to extend its applicability.

• Expand slightly on rural vs. urban rehabilitation differences.

Response: We have reported in the discussion section that “Within Pakistan, urban centers often provide better access to rehabilitation facilities and technologies (e.g., robotic gait training), while rural areas face barriers like transportation and limited therapist availability. This disparity may affect the applicability of the 33 meters MCID, as rural patients may have less frequent therapy sessions.”

Minor Comments

Language and Grammar

There are several small grammar errors throughout the manuscript. A careful language edit is recommended.

Response: The clean version of the revised manuscript has undergone a thorough review to correct minor grammatical errors, such as ensuring consistency in grammar and refining transitions between the results and discussion sections.

Consistency

• Use either 2MWT or 2WMT consistently.

• Keep formatting consistent for ABC_Gait.

• Use either “33 m” or “33 meters” consistently.

Response: We have put this into consideration while making the required amendment to ensure consistency.

Tables and Repetition

• Some subgroup results are repeated in multiple sections. Remove repetition.

Response: Repetitions have been removed in the clean version of the revised manuscript.

• Table formatting can be simplified.

Response: Table formatting has been simplified.

Recommendation

Major Revision

The study is valuable and relevant, but clarification and careful editing are needed before publication.

---

## [Editor Report · Decision Letter 3]

26 Mar 2026

Validating the 2-minute walk test MCID for subacute stroke patients: A Pakistani multicenter cohort analysis

PONE-D-25-43400R3

Dear Dr. Edward Muteesasira,

We’re pleased to inform you that your manuscript has been judged scientifically suitable for publication and will be formally accepted for publication once it meets all outstanding technical requirements.

Kind regards,

Sohel Ahmed, BPT, MPT, MDMR

Academic Editor

PLOS One
---

## [Editor Report · Acceptance letter]

PONE-D-25-43400R3

PLOS One

Dear Dr. Muteesasira,

I'm pleased to inform you that your manuscript has been deemed suitable for publication in PLOS One. Congratulations! Your manuscript is now being handed over to our production team.

Kind regards,

on behalf of

Dr. Sohel Ahmed

Academic Editor

PLOS One